# Social Distancing among Medical Students during the 2019 Coronavirus Disease Pandemic in China: Disease Awareness, Anxiety Disorder, Depression, and Behavioral Activities

**DOI:** 10.3390/ijerph17145047

**Published:** 2020-07-14

**Authors:** Huidi Xiao, Wen Shu, Menglong Li, Ziang Li, Fangbiao Tao, Xiaoyan Wu, Yizhen Yu, Heng Meng, Sten H. Vermund, Yifei Hu

**Affiliations:** 1Department of Child and Adolescent Health and Maternal Care, School of Public Health, Capital Medical University, Beijing 100069, China; xhd19988023@163.com (H.X.); nmg_sw026@163.com (W.S.); limenglong_ph@163.com (M.L.); 15622764530@163.com (Z.L.); 2Department of Maternal, Child and Adolescent Health, School of Public Health, Anhui Medical University, Hefei 230032, China; fbtao@ahmu.edu.cn (F.T.); xywu85@126.com (X.W.); 3Department of Child and Women Health Care, School of Public Health, Tongji Medical College, Huazhong University of Science and Technology, Wuhan 430030, China; yuyizhen650@163.com (Y.Y.); menghengmay@hust.edu.cn (H.M.); 4Department of Epidemiology of Microbial Diseases, Yale School of Public Health, New Haven, CT 06520, USA; sten.vermund@yale.edu; 5Department of Pediatrics, Yale School of Medicine, New Haven, CT 06520, USA

**Keywords:** COVID-19, anxiety disorder, depression, social distancing, personal protective measures, medical students, knowledge, China

## Abstract

Background: During the coronavirus disease (COVID-19) pandemic, harsh social distancing measures were taken in China to contain viral spread. We examined their impact on the lives of medical students. Methods: A nation-wide cross-sectional survey of college students was conducted from 4–12 February 2020. We enrolled medical students studying public health in Beijing and Wuhan to assess their COVID-19 awareness and to evaluate their mental health status/behaviors using a self-administered questionnaire. We used the Patient Generalized Anxiety Disorder-7 and Health Questionnaire-9 to measure anxiety disorders and depression. We used multivariable logistic regression and path analysis to assess the associations between covariates and anxiety disorder/depression. Results: Of 933 students, 898 (96.2%) reported wearing masks frequently when going out, 723 (77.5%) reported daily handwashing with soap, 676 (72.5%) washed hands immediately after arriving home, and 914 (98.0%) reported staying home as much as possible. Prevalence of anxiety disorder was 17.1% and depression was 25.3%. Multivariable logistic regression showed anxiety to be associated with graduate student status (odds ratio (aOR) = 2.0; 95% confidence interval (CI): 1.2–3.5), negative thoughts or actions (aOR = 1.6; 95% CI: 1.4–1.7), and feeling depressed (aOR = 6.8; 95% CI: 4.0–11.7). Beijing students were significantly less likely to have anxiety than those in the Wuhan epicenter (aOR = 0.9; 95% CI: 0.8–1.0), but depression did not differ. Depression was associated with female students (aOR = 2.0; 95% CI: 1.2–3.3), negative thoughts or actions (aOR = 1.7; 95% CI: 1.5–1.9), and anxiety disorder (aOR = 5.8; 95% CI: 3.4–9.9). Path analysis validated these same predictors. Conclusions: Despite medical students’ knowledge of disease control and prevention, their lives were greatly affected by social distancing, especially in the Wuhan epicenter. Even well-informed students needed psychological support during these extraordinarily stressful times.

## 1. Introduction

Coronavirus disease (COVID-19) is a public health emergency of worldwide concern [1]. Up through June 2020, COVID-19 had affected 213 countries, territories or areas, and two international conveyances (cruise ships) [2] since its initial December 2019 report as an “unknown pneumonia” in Wuhan, Hubei Province, China [3]. Infections and deaths increased rapidly with global travel fueling transmission spread worldwide, first to other parts of China, South Korea, and Iran, and soon thereafter to Europe and the United States. Other nations such as Russia and Brazil then experienced expanding case numbers while infections in Africa have also been mounting. 

To interrupt further transmission, many Chinese provinces and cities suspended public transportation [4] and even locked down cities altogether. These were unprecedented and harsh social distancing measures and they affected every aspect of daily life in China, as elsewhere.

“Social distancing” is done to reduce close physical interactions (<2 m) between people to avoid viral transmission. Chinese measures included closure of schools, office buildings, and public markets; events were cancelled, and gatherings discouraged [5]. The United Nations Educational, Scientific and Cultural Organization (UNESCO) estimates that nationwide closures in more than 160 countries have affected over 87% of students in the world, with wider closures affecting even more [6]. A 2009 influenza A (H1N1) pandemic survey in Hong Kong demonstrated marked anxiety associated with distancing [7]. A 2020 COVID-19 survey in China found that many respondents (including students) reported symptoms of moderate to severe anxiety and depression [8]. 

Medical students are a reservoir for the future health labor force. In the early response in the epicenter in China, as well as other countries, the sudden outbreak overwhelmed health professional’s preparedness in terms of personal protective equipment (PPE) shortages and psychological readiness [9,10]. It is critical to understand how much the pandemic affects this future health labor force. We sought to understand the psychological effects of distancing measures and possible effects on medical student wellness. Even without pandemic stresses, medical students may have some anxiety and depression due to their high study-related or job-seeking burdens [11,12,13]. We studied the associations between COVID-19 distancing and the lives of public health students on anxiety [14], depression, and other behaviors. We also examined the differences among public health students in two universities, Capital Medical University (CCMU) in Beijing, and Huazhong University of Science and Technology (HUST) in Wuhan, the early epicenter of the COVID-19 outbreak in China.

## 2. Methods

### 2.1. Setting, Design, and Questionnaire

We conducted a cross-sectional survey among the medical students in two schools of public health at two universities: CCMU in Beijing and HUST in Wuhan. The present study is part of a nation-wide cross-sectional survey of college students conducted from 4–12 February 2020. We developed a self-administrated, 84-item questionnaire focused on: (1) COVID-19 awareness; (2) personal protective measures; (3) mental health status; and (4) behavioral changes in the past 14 days. Our instrument was based on social-cognitive theory and the Theory of Planned Behavior and we measured knowledge, attitude, belief, and practice (KABP) [15], enhanced with psychological assessments. We used the validated Generalized Anxiety Disorder-7 (GAD-7) and Patient Health Questionnaire-9 (PHQ-9) to assess anxiety disorder and depression, respectively [16,17]. Most questions were one-choice, brief, and easy to answer, such that it took only 10 min on average to complete the survey. We used Wenjuanxing^®^ (Changsha Haoxing Information Technology Co., Ltd., Changsha, China) software for the informed consent process and the online survey questionnaire. Participants obtained and completed the self-administrated questionnaire on the mobile phone by scanning the QR code using WeChat^®^ software. After consent was obtained, we detailed the workflow of the survey with one page of text. The investigators received the completed questionnaires via the software platform.

### 2.2. Participants, Recruitment, Covariates, and Outcomes

Facilitated by their teachers, we approached all medical students with public health majors in the CCMU and HUST schools of public health. The study was approved by Ethics Review Board of Capital Medical University (2020SY004) and Anhui Medical University (20200319). Informed consent and a response to the questionnaire were obtained from 933 of 1061 (87.9%) students who were approached. Hence, we could infer the different preventive knowledge, behaviors, and psychological status modified by distance to the epicenter amid social distancing in the pandemic.

Covariates included socio-demographic characteristics, knowledge of COVID-19, personal protective measure, behaviors, and degree of worry about the virus. Variables included sex, age, year of study, university, living quarters, knowledge about the COVID-19 incubation period, mortality, susceptibility (e.g., chronic illness), drugs taken, mask wearing, face and hand hygiene, times and reasons for going out, room cleanliness and ventilation via windows, and concerns about the COVID-19 epidemic and contracting the virus. We also asked about negative thoughts or actions (“always feel dirty”, “feel uneasy in a crowded place”, “often suspect being infected”, “worse appetite than before”, “feel less energic than before”, “hold unhappy intentions in my heart” and “angry with others when in a bad mood”), positive thoughts or actions (“accept the truth when facing an obstacle” and “relieve pain in a positive way”), healthy lifestyles (“work and rest regularly”, “arise regularly”, “sleep regularly” and “have meals regularly”), video screen time per day, and the number of anger episodes or quarrels in the past week.

Principal outcomes were anxiety disorder and depression measured by scale scores of GAD-7 and PHQ-9, respectively. The GAD-7 scale score was divided into four categories: normal (0–4), mild (5–9), moderate (10–14), and severe (15–21). The PHQ-9 scale score was divided into five categories: normal (0–4), mild (5–9), moderate (10–14), moderate to severe (15–19), and severe (20–27). If a participant’s score was ≥5 points (i.e., mild or above), we considered the student to have evidence of anxiety disorder.

### 2.3. Statistical Analysis

We used descriptive statistics including the Chi-squared test for the associations of COVID-19 knowledge with sociodemographic characteristics and use of personal protective measures. We used multivariable logistic regression to examine the association between independent variables and covariates with anxiety disorder and/or depression. We deployed the Hosmer–Lemeshow test to determine the goodness-of-fit of the logistic regression model. Only variables with two-sided *p* ≤ 0.05 were deemed significant in the final model. We used path analysis to determine the interplay of covariates with anxiety disorder and depression via a structural equation model. We used maximum likelihood estimation and assessed the goodness-of-fit by absolute fit indices that determine how well the a priori model predicts the actual data, including the root mean square error of approximation (RMSEA), goodness of fit index (GFI), and adjusted goodness of fit index (AGFI). We also used incremental or relative fit indices, specifically the incremental fit indices (IFI), comparative fit indices (CFI), normed fit index (NFI), and the non-normed fit Tucker-Lewis index (TLI). RMSEA < 0.10 and GFI and AGFI > 0.90 indicate the model fits well. The incremental fit measures—CFI, NFI, IFI, and TLI—are >0.90 when the model fits well. We used SPSS Statistic^®^ 21.0 and SPSS Amos^®^ 26.0 Graphics software (IBM SPSS Statistics, New York, NY, USA).

## 3. Results

Of 933 participated students, 558 students attended CCMU (94.4% agreed to participate) in Beijing, and 375 students attended HUST (79.6% agreed) in Wuhan.

### 3.1. Preventive Knowledge, Anxiety Disorders, and Depression

Graduate students doing Masters of Public Health degrees did better than undergraduates in the knowledge questions that were answered correctly (*p* = 0.037). It is important to note that the Chinese system has undergraduate medical students and before graduation, medical students can pursue specialties such as public health, pediatrics, etc.; then at the graduate level they may choose to major in epidemiology and statistics, environmental health, etc., if they chose the public health specialty. The prevalence of anxiety disorder (*p* = 0.015) and depression (*p* < 0.001) in women was significantly higher than in men. Anxiety disorder was higher in Wuhan students than CCMU students (*p* = 0.001; Table 1).

The correct answers to the four knowledge questions were given by 97.2%, 67.1%, 69.3%, and 89.4% of respondents (Appendix A). The awareness of mortality risk was higher in HUST students than CCMU students (*p* = 0.04).

### 3.2. Comparing Protective Behaviors between the Two Universities

As to preventive measures and behaviors during social distancing, 898 (96.2%) reported wearing masks frequently when going out, 869 (93.1%) washed their hands with water regularly, 723 (77.5%) washed their hands with soap every day, 676 (72.5%) washed their hands immediately after arriving home, 239 (25.6%) considered it difficult to wash their hands for at least 20 s, 295 (31.6%) washed their hands for more than 20 s frequently, 914 (98.0%) avoided unnecessary outings (i.e., they tried to stay at home as much as possible), and 878 (94.1%) kept clean, well-ventilated rooms. Beijing-based CCMU students were significantly more likely to report wearing masks (*p* = 0.037), avoiding touching their mouths, noses, and eyes with their hands (*p* < 0.001), washing their hands immediately after arriving home (*p* < 0.001), and handwashing for at least 20 s (*p* < 0.001). Wuhan-based students at HUST were more likely to report washing their hands with soap (*p* < 0.001), staying at home (*p* = 0.028), and keeping their rooms clean and well-ventilated (*p* = 0.044; Table 2).

### 3.3. Comparing Anxiety Disorder and Depression Prevalence between the Two Universities

Assessing anxiety disorder, 773 (82.9%) were classified as normal, 117 (12.5%) had mild anxiety, 30 (3.2%) had moderate anxiety, and 13 (1.4%) had severe anxiety disorder. Assessing depression, 697 (74.7%) students were classified as normal, 165 (17.7%) had mild depression, 43 (4.6%) had moderate depression, 18 (1.9%) had moderate to severe depression, and 10 (1.1%) had severe depression. The prevalence of anxiety disorder differed between the two universities, and was significantly higher in Wuhan (*p* = 0.001) which was far more severely affected by COVID-19. The prevalence of depression between the two universities was also higher in Wuhan, but this may have been due to chance (*p* = 0.12; Table 1 and Figure 1).

Bar charts present the distribution of different degrees of anxiety disorder and depression, comparing students at Capital Medical University (CCMU) in Beijing with students at Huazhong University of Science and Technology (HUST) in Wuhan. The *X*-axis represents the different degrees of anxiety disorder and depression, and the *Y*-axis represents the proportion of students.

### 3.4. Predictors of Anxiety Disorder and Depression Using Multivariable Logistic Regression and Path Analysis

Multivariable logistic regression shows that being a graduate student (Adjusted odds ratio (aOR) = 2.03; 95% confidence interval (CI): 1.18–3.49; *p* = 0.011), having negative thoughts or actions (aOR = 1.55; 95% CI: 1.38–1.73; *p* < 0.001), and feeling depressed (aOR = 6.84; 95% CI: 4.00–11.71; *p* < 0.001) were associated with a higher likelihood of anxiety. Students at CCMU, far from the Wuhan epicenter, were less likely to experience anxiety (aOR = 0.90; 95% CI: 0.82–1.00; *p* = 0.049; Table 3). Women students (aOR = 1.98; 95% CI: 1.19–3.29; *p* = 0.009), persons having negative thoughts or actions (aOR 1.68; 95% CI: 1.50–1.88; *p* < 0.001), and persons with anxiety (aOR = 5.81; 95% CI: 3.43–9.86; *p* < 0.001) had higher odds of having some depression. Having a healthy lifestyle was associated with less depression (aOR = 0.88; 95% CI: 0.79–0.97; *p* = 0.013).

Across the two sites, 426 (48.7%) students reported using computers or other electronic devices over 4 h daily, 484 (51.9%) used their cellphones over 4 h daily, 635 (68.1%) woke up later than usual, 234 (24.0%) went to bed later than usual, 234 (35.1%) worked and rested irregularly, 157 (16.8%) had meals irregularly, and 221 (23.7%) ate different volumes of food from usual (either more or less). Within one week before the survey, 306 (32.8%) reported having gotten angry at others, 201 (21.5%) reported one or more quarrels, 585 (62.7%) felt terrible because of the epidemic, and 112 (12.0%) quarreled with others online. 

Figure 2 shows the factors relevant to anxiety disorder and depression, and Appendix A presents standardized estimation of coefficient values. Students in their senior grade year (β = 0.074) suffered more from anxiety disorders. Negative thoughts or actions were associated with depression (β = 0.86) while healthy lifestyles (β = −0.077) were negatively associated with depression. Longer video screen time (β = −0.24) negatively affected a healthy lifestyle. Concerns about the COVID-19 epidemic were associated with more negative thoughts or actions (β = 0.23) and anger and quarreling behaviors (β = 0.20); negativity and anger/quarrels were correlated (β = 0.34). Overall, negative thoughts or actions predicted a higher impact of both anxiety disorders (β = 0.87) and depression (β = 0.86). Goodness-of-fit indices for the model were good and key parameters were RMSEA = 0.051, GFI = 0.93, AGFI = 0.91, CFI = 0.93, NFI = 0.91, IFI = 0.93, and TLI = 0.92.

## 4. Discussion

Our survey findings suggest that the awareness of medical students of COVID-19 fundamentals was very high, as might be expected given the profile in Chinese society by February 2020. Anxiety and depression were common among medical students in whom social distancing was reported with longer video screen time and less healthy lifestyles. Concern about the epidemic was associated with negative actions and thoughts, which were, in turn, associated with an increased likelihood of anxiety disorders. The HUST medical students in Wuhan, the epicenter, presented with higher anxiety than did CCMU medical students in Beijing. The findings underscore our study’s significance that the health reservoir suffer from psychological stress and need attention.

Less than half (43.7%) of the medical students had fully correct (four out of four questions) knowledge of the COVID-19 epidemic situation. Students, all of whom were studying public health, were less knowledgeable about mortality and susceptible groups, compared to incubation period and available drugs (Appendix A). Over 90% of students complied with social distancing and effective preventive measures, like wearing masks frequently when going out, washing hands with water regularly, avoiding unnecessary outings (i.e., staying at home), and keeping their rooms clean and ventilated [18]. According to our findings, students preferred washing their hands with only water to also using soap; about half (54%) of students tried to avoid touching their mouth, nose, and eyes with their hands, though both behaviors can reduce the risk of exposure [19]. A quarter (25.6%) of the students considered it hard to wash their hands for at least 20 s and less than a third (31.6%) washed their hands over 20 s frequently. Given that good hand hygiene can effectively prevent virus transmission [20] and washing hands for at least 20 s is a basic component of hand hygiene as per the World Health Organization (WHO, Geneva, Switzerland), it was disappointing to learn from medical students that they found this challenging [21]. 

It is likely that, comparing the two participating universities, notable differences were fueled by the differing epidemiologic context of the two urban venues. Among the 44.0% of students from HUST who lived in the epicenter of Hubei province or adjacent provinces, they were more likely to obey harsher social distancing rules, like staying at home and deploying hand hygiene. While most students (88.2%) from CCMU lived far away from Hubei province, they reported being more fastidious in wearing masks.

During social distancing, our study showed 17.1% of students had anxiety disorder symptoms (mainly mild), and 25.3% of students had depression symptoms (mainly mild). An interesting survey in 190 Chinese cities surveyed the general population twice: during the initial outbreak and during the epidemic’s peak four weeks later [22]. They surveyed demographics, symptoms, knowledge, concerns, and precautionary measures against COVID-19. Among the 333 persons who took both surveys, post-traumatic stress disorder (PTSD by the mean Impact of Event Scale-Revised (IES-R) scale scores) declined with time, but at both surveys, the mean IES-R scores of the first- and second-survey respondents were above the cut-off scores (>24) for PTSD symptoms, suggesting that the reduction in scores was not of clinical significance [22]. Previous studies report that student populations can be more vulnerable towards stress-related anxiety and depression [22,23]. Our findings underscore the importance of providing essential psychological support to students, even when they are as well-informed as are medical students in a public health track. 

Both multivariable logistic regression and path analysis reinforced the findings of principal factors related to anxiety disorder and depression. Compared to undergraduates, graduate students had a higher risk of anxiety [13,24]. We think that this may be due to increased pressure for job-seeking or completion of thesis required for graduation. Females were more likely to be depressed compared to men, consistent with prior studies [24,25]. Healthy lifestyles were negatively associated with depression, compatible with other findings that healthy lifestyles can improve mental health [26]. Some students adopted an unhealthy lifestyle during social distancing, reflecting difficulties in adjusting to domestic life for a prolonged time; as expected, many students reported bad moods and/or behaviors [27]. Nearly a quarter of students (23.7%) changed their diets by eating more or less than usual. Either excessive or poor appetites can be symptoms of depression [28]. Longer video screen time had an indirect impact on depression, as other studies reported [29,30]. Half of our participants spent over four hours on electronic devices; aside from increasing risk for depression, this can negatively affect vision, or spawn weight gain and cardiovascular risk from adoption of a sedentary life [31]. 

We found that the prevalence of anxiety disorder was higher in Wuhan than in Beijing (*p* = 0.001), while depression was also somewhat higher (*p* = 0.12). Research on college students from Changzhi Medical College (Shanxi Province, about midway between Wuhan and Beijing) in the same time period reported that 24.9% of students had anxiety disorder using screening criteria similar to ours [32]. It is unclear whether differences in reported prevalence are related to school location, though it is plausible that proximity to the pandemic’s epicenter would be more anxiety-provoking [33]. That negative thoughts or actions, anger and quarreling behaviors, and concerns about COVID-19 were all positively related to either depression or anxiety disorder has been seen in other studies [32,34,35]. Many of our participants reported that they had been angry and quarreling within one week before the survey; 46.7% of them felt terrible because of COVID-19. It is plausible that irritable behaviors suggest that mental status or moods may have been affected by social distancing due to lack of normal social activities during the pandemic [36]. As expected, we found a strong positive interaction effect between anxiety disorder and depression. Reduced physical activity may increase anxiety or depression, but we did not find a clear association in our survey.

Strengths of the study include its uniqueness, a survey in two sites (including Wuhan) conducted just two months after the report of the pulmonary syndrome and one month after recognition of SARS-CoV-2 circulation in China. Limitations include the cross-sectional survey design. While we could compare sites and assess predictors of anxiety and depression, we could not assess mental health circumstances before the emergence of the virus and therefore cannot infer temporality, the vital element to assess causality. The participants are medical students studying public health from two universities; therefore, results and conclusions cannot be generalized to other populations. Moreover, because we used self-rating scales, the frequency of anxiety disorder and depression symptoms self-reported by the students is less reliable than thorough clinical diagnoses. As an emerging disease, our understanding of COVID-19 keeps evolving and we selected “knowledge questions” based on what was known in late January 2020, based on several rounds of expert consultations and consistent with WHO and US Centers for Disease Control and Prevention updates.

The COVID-19 pandemic has posed an unprecedented impact on the lives of medical students. Necessary psychological support was not available to them during the time of social distancing beyond an awareness campaign regarding preventive measures. It is likely that more economically vulnerable persons would have stress levels exceeding that of medical students, but we learned that this well-informed population was nonetheless in substantial distress in the context of epidemic concerns. Mental health pressures in medical students are likely compounded among practicing clinicians; these professionals need psychosocial assistance at the time of pandemic mental stress. We recommend incorporating pandemic preparedness education within health education, including mental health elements, especially within the healthcare labor force. 

## Figures and Tables

**Figure 1 ijerph-17-05047-f001:**
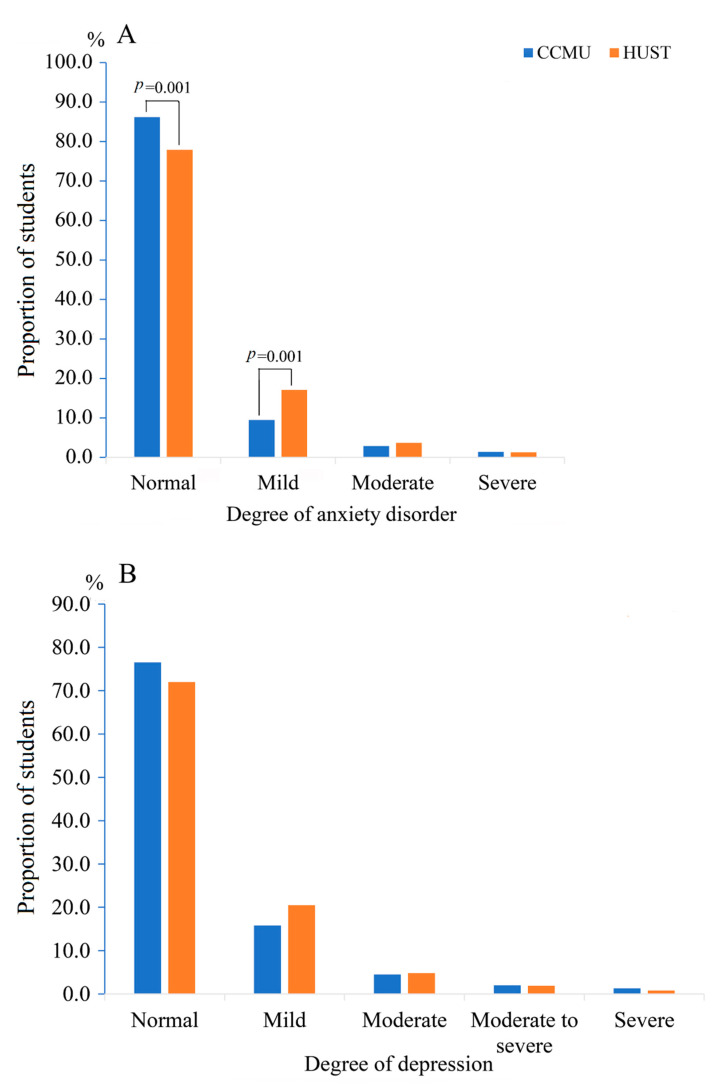
Distribution of different degrees of (**A**) anxiety disorder and (**B**) depression at two universities.

**Figure 2 ijerph-17-05047-f002:**
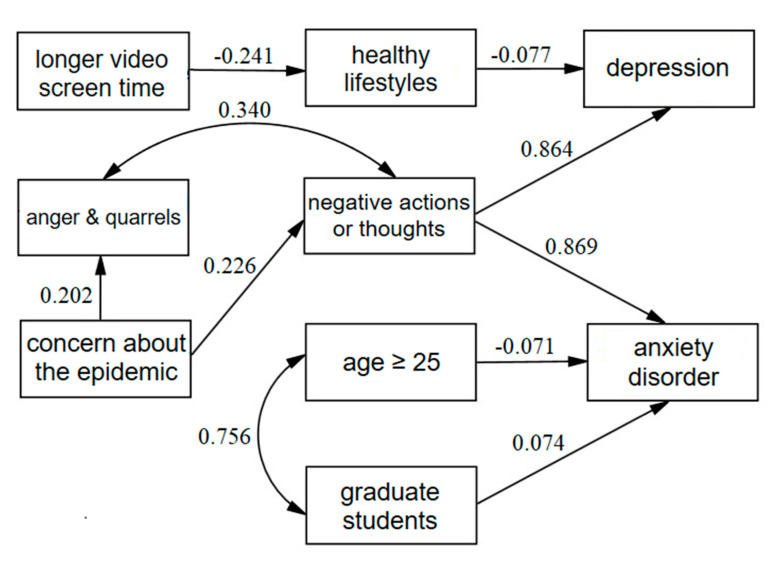
Relationship between anxiety disorder, depression, and relevant factors using path analysis. Path analysis establishes a structural equation model between anxiety disorder, depression, and influencing factors. Arrows represent the direction of correlation between variables, and numbers represent coefficient values.

**Table 1 ijerph-17-05047-t001:** Coronavirus disease (COVID-19) preventive knowledge, anxiety disorder, and depression of college students categorized by socio-demographic characteristics (*N* = 933).

Characteristics	*n* (%)	Knowledge			Anxiety Disorder		Depression	
		Score (0–4)	Correct Answers (%)	*p*	Prevalence (%)	*p*	Prevalence (%)	*p*
Sex				0.175		0.015		<0.001
Male	279 (29.9)	3.12	218 (78.0)		35 (12.5)		47 (16.8)	
Female	654 (70.1)	3.28	536 (82.0)		125 (19.1)		189 (28.9)	
Age in years				0.193		0.58		0.335
17–24	755 (80.9)	3.2	604 (80.0)		132 (17.5)		196 (26.0)	
25 or older	178 (19.1)	3.37	150 (84.3)		28 (15.7)		40 (22.5)	
Grade of medical students				0.089		0.3		0.57
Undergraduates	620 (66.5)	3.15	488 (78.8)		98 (15.8)		154 (24.8)	
MPH graduate students	228 (24.4)	3.4	194 (85.0)		46 (20.2)		63 (27.6)	
DrPH graduate students	85 (9.1）	3.34	71 (83.5)		16 (18.8)		19 (22.4)	
University				0.34		0.001		0.12
CCMU	558 (59.8)	3.21	448 (80.3)		77 (13.8)		131 (23.5)	
HUST	375 (40.2)	3.26	306 (81.5)		83 (22.1)		105 (28.0)	
Living residence				0.061		0.073		0.25
Village	350 (37.5)	3.16	277 (79.0)		70 (20.0)		96 (27.4)	
City	583 (62.5)	3.27	477 (81.8)		90 (15.4)		150 (24.0)	
Overall		3.23	753 (80.8)		160 (17.1)		236 (25.3)	

CCMC = Capital Medical University; HUST = Huazhong University of Science and Technology. MPH = Masters of Public Health degree; DrPH = Doctor of Public Health degree.

**Table 2 ijerph-17-05047-t002:** Personal protective behaviors between students at two universities in Beijing and Wuhan (*N* = 933).

Behaviors	Levels	*N* (%)			χ^2^	*p*
		Overall	CCMU (*n* = 558)	HUST (*n* = 375)		
Wearing masks when going out	Frequently	898 (96.2)	543 (97.3)	355 (94.7)	4.35	0.037
	Sometimes	22 (2.4)	8 (1.4)	14 (3.7)		
	Hardly ever	13 (1.4)	7 (1.3)	6 (1.6)		
Avoiding touching mouth, nose, and eyes with hands	Frequently	504 (54)	331 (59.3)	173 (46.1)	15.70	<0.001
	Sometimes	306 (32.8)	164 (29.4)	142 (37.9)		
	Hardly ever	123 (13.2)	63 (11.3)	60 (16.0)		
Times of hands washing with water per day	0–2	64 (6.9)	36 (6.5)	28 (7.5)	0.36	0.55
	3–5	397 (42.6)	233 (41.8)	164 (43.7)		
	6–10	364 (39.0)	217 (38.9)	147 (39.2)		
	11–15	66 (7.1)	42 (7.5)	24 (6.4)		
	>15	42 (4.5)	30 (5.4)	12 (3.2)		
Times of hands washing with soap per day	0–2	210 (22.5)	95 (17.0)	115 (30.7)	23.93	<0.001
	3–5	419 (44.9)	257 (46.1)	162 (43.2)		
	6–10	241 (25.8)	156 (28.0)	85 (22.7)		
	11–15	35 (3.8)	27 (4.8)	8 (2.1)		
	>15	28 (3.0)	23 (4.1)	3 (1.3)		
Washing hands immediately after arriving home	Frequently	676 (72.5)	443 (79.4)	233 (62.1)	33.47	<0.001
	Sometimes	181 (19.4)	85 (15.2)	96 (25.6)		
	Hardly ever	76 (8.1)	30 (5.4)	46 (12.3)		
Washing hands for 20 s each time	Frequently	295 (31.6)	215 (38.5)	89 (21.3)	30.68	<0.001
	Sometimes	399 (42.8)	235 (42.1)	164 (43.7)		
	Hardly ever	239 (25.6)	108 (19.4)	131 (34.9)		
Times of going out per day	0−2	914 (98.0)	542 (97.1)	372 (99.2)	4.81	0.028
	3−4	13 (1.4)	10 (1.8)	3 (0.8)		
	5 or more	6 (0.6)	6 (1.1)	0		
Keeping room clean and window ventilated	Frequently	878 (94.1)	518 (92.8)	360 (96.0)	4.06	0.044
	Sometimes	43 (4.6)	30 (5.4)	13 (3.5)		
	Hardly ever	12 (1.3)	10 (1.8)	2 (0.5)		

Note: Chi-squared analysis performed only between the first row of each item of the two universities. CCMC = Capital Medical University (CCMU) in Beijing; HUST = Huazhong University of Science and Technology (HUST) in Wuhan.

**Table 3 ijerph-17-05047-t003:** Predictors of depression and anxiety disorder using multivariable logistic regression among medical students of Capital Medical University (CCMU) in Beijing and Huazhong University of Science and Technology (HUST) in Wuhan (*N* = 933).

Dependent Variable	Independent Variable	Adjusted Odds Ratio (95% CI)	*p*
Anxiety disorder	Sex (ref = male)	1.09 (0.60−1.97)	0.79
(yes or no)	Age (ref = age ≤ 24)	0.37 (0.14−0.95)	0.04
	Grade (ref = undergraduates)	2.03 (1.18−3.49)	0.011
	University/site (ref = HUST in Wuhan)	0.90 (0.82−1.00)	0.049
	Concerns about the COVID-19 epidemic	1.07 (0.90−1.27)	0.45
	Concerns about contracting the COVID-19	1.01 (0.90−1.13)	0.85
	Negative thoughts or actions ^a^	1.55 (1.38−1.73)	<0.001
	Positive thoughts or actions ^b^	1.02 (0.87−1.20)	0.8
	Healthy lifestyles ^c^	1.04 (0.92−1.16)	0.56
	Anger and quarreling behaviors	1.05 (0.95−1.16)	0.31
	Less going out	1.13 (0.89−1.46)	0.32
	Depression	6.84 (4.00−11.71)	<0.001
Depression	Sex (ref = male)	1.98 (1.19−3.29)	0.009
(yes or no)	Concerns about the COVID-19 epidemic	1.11 (0.96−1.28)	0.18
	Concerns about contracting the COVID-19	0.97 (0.88−1.07)	0.5
	Negative thoughts or actions ^a^	1.68 (1.50−1.88)	<0.001
	Positive thoughts or actions ^b^	0.93 (0.81−1.06)	0.27
	Longer video screen time	1.05 (0.95−1.15)	0.37
	Healthy lifestyles ^c^	0.88 (0.79−0.97)	0.01
	Anger and quarreling behaviors	1.08 (0.98−1.18)	0.11
	Anxiety disorder	5.81 (3.43−9.86)	<0.001

^a^ includes: “always feel dirty”, “feel uneasy in a crowded place”, “often suspect being infected”, “worse appetite than before”, “feel less energic than before”, “hold unhappy intentions in my heart”, and “angry with others when in a bad mood”. ^b^ includes: “accept the truth when facing obstacles”, and “relieve pain in a positive way”. ^c^ includes: “work and rest regularly”, “arise regularly”, “sleep regularly”, and “have meals regularly”.

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
