# Peer review of "Social Distancing among Medical Students during the 2019 Coronavirus Disease Pandemic in China: Disease Awareness, Anxiety Disorder, Depression, and Behavioral Activities"

_ijerph, 2020, doi:10.3390/ijerph17145047_

Round 1
Reviewer 1 Report
Revision of the manuscript IJERPH-838567
This study investigated the impact of COVID-19 distancing on the lives of medical students specializing in public health on depression, anxiety and other behavioral measures using a cross-sectional design. The study is well conducted, has an appropriate design and methodology, and is well presented. I would congratulate the Authors for their work as covers an important aspect of the impact of COVID-19, such as metal status of medical students. I have only few minor comments that I hope will be useful to improve the scientific quality of the manuscript.
- Within behavioral measures, did the Authors think to assess also physical activity? It is possible that an association exists between reduced physical activity and increased anxiety and depression. Please elaborate this point in the Discussion section.
- Was the academic achievement considered? I mean as possible covariates.
- Please check the consistency of the verbal tenses throughout the manuscript.
- Please check minor typos throughout the manuscript (e.g., last line of the first paragraph of discussion: “that” should be “than”).
Author Response
Reviewer #1
1.Within behavioral measures, did the Authors think to assess also physical activity? It is possible that an association exists between reduced physical activity and increased anxiety and depression. Please elaborate this point in the Discussion section.
Response: Thank you for helpful comments. While we assessed physical activity, we did not observe an association between physical activity with anxiety or depression. We still highlight it in the discussion section, line 26-27, page 21 as” Reduced physical activity may increase anxiety or depression, but our study did not find such association.”.
2.Was the academic achievement considered? I mean as possible covariates.
Response: Thank you for the comments. We considered it as one of the covariates as “graduate students” or not.
3.Please check the consistency of the verb tenses throughout the manuscript.
Response: Thank you for the reminder. We went through the whole manuscript and revised it as appropriate. One of our co-authors Dr Sten H.Vermund is a native English speaker, who reviewed and revised the whole paper to ensure the accuracy of the description.
4.Please check minor typos throughout the manuscript (e.g., last line of the first paragraph of discussion: "that" should be "than").
Response: Thank you for the reminder and we have changed it accordingly
Reviewer 2 Report
This study compares anxiety and depression among medical students between a university at the epicentre of COVID and a university distant from the epicentre. The manuscript is timely and well-written. I have a number of minor suggestions for revision:
Line 67: “Medical students are a reservoir the future health labor force.” – change to “Medical students are a reservoir for the future health labor force.”
Page 10, just before Figure 1: “while the prevalence of depression was higher in Wuhan, this may have been due to chance” – this sentence implies there may have been a statistical difference between the universities, but according to Table 1, this was not statistically significant. Please revise.
Figure 1: The resolution of this figure can be improved. Since figures should “stand alone” from the text, please indicate in the figure which university is from Beijing and which is from Wuhan. If there are any statistical differences between the universities, this should be indicated by symbols on the figure.
Discussion, first paragraph, last sentence: “The HUST medical students in Wuhan, the epicenter, presented with higher depression and anxiety that did CCMU medical students in Beijing.” While this is true of anxiety, according to table 1, the p-value for difference in depression was 0.12, which is not statistically significant. I suggest modifying this sentence to better reflect your results.
Bottom of page 17: “The participants were medical students now studying public health students in two universities…” This sentence needs some re-wording.
Author Response
Reviewer #2
1.Line 67: "Medical students are a reservoir the future health labor force." - change to "Medical students are a reservoir for the future health labor force."
Response: Thank you for the helpful revision and we have revised it accordingly.
2.Page 10, just before Figure 1: "while the prevalence of depression was higher in Wuhan, this may have been due to chance" - this sentence implies there may have been a statistical difference between the universities, but according to Table 1, this was not statistically significant. Please revise.
Response: Thank you for the reminder and we revised the sentence and moved the misplaced half sentence to the end of former sentence where it belonged.
3.Figure 1: The resolution of this figure can be improved. Since figures should "stand alone" from the text, please indicate in the figure which university is from Beijing and which is from Wuhan. If there are any statistical differences between the universities, this should be indicated by symbols on the figure.
Response: Thank you for this advice. We have improved the resolution of this figure and have indicated the statistical differences. We had indicated the university and city in the figure and caption.
4.Discussion, first paragraph, last sentence: "The HUST medical students in Wuhan, the epicenter, presented with higher depression and anxiety that did CCMU medical students in Beijing." While this is true of anxiety, according to table 1, the p-value for difference in depression was 0.12, which is not statistically significant. I suggest modifying this sentence to better reflect your results.
Response: Thank you for your critical comments; we revised the sentence.
5.Bottom of page 17: "The participants were medical students now studying public health students in two universities..." This sentence needs some re-wording
Response: Thanks for the reminder; we revised the sentence.
Reviewer 3 Report
Social Distancing among Medical Students during the 2019 Coronavirus Disease Pandemic in China: Disease Awareness, Depression, Anxiety, and Behavioral Activities
Referee Report
This paper conducts a survey of college medical students enrolled in public health to assess their COVID-19 awareness and to evaluate their mental health status and related behavior. In particular, first, it describes preventive knowledge, anxiety, and depression, as well as personal protective behavior. Second, it estimates the main predictors of depression and anxiety disorders. The study concludes that, even though students have high knowledge of prevention for COVID-19, they still are psychologically affected by the epidemic, especially in Wuhan.
Overall, the article has potential and reports relevant evidence. The fact that even medical students well knowledgeable of the epidemic are mentally affected is of policy relevance. The results suggest that there is a need also for students in the medical setting to be supported during health shocks. The article is very concise, and lack details on the methods and more background. The data are novel and interesting and the policy implications of this research could be better explained. I do have some suggestions for the authors to make the article more consistent and clearer.
Major comments:
Unfortunately, the manuscript did not have row numbers in all pages and did not have a page number, so I apologize if the references to the text are not more precise
- Purpose of the paper/outcomes
From the reading of the entire manuscript, the purpose of the paper appears the following:
(i) to describe Covid-19 awareness on preventive behavior, anxiety and depression (Table 1)
(ii) to describe personal protective behavior (Table 2)
(iii) to look at predictors of anxiety and depression (Table 3)
(iv) to further understand the relationship between depression and anxiety and their predictors (Figure 2)
Both the abstract and the introduction describes the plan for the analysis, but there is inconsistency about the goals of the analysis. For example, the abstract describes (in the methods) the descriptive part, but not the assessment of the associations between covariates and depression/anxiety (even though results on those are described in results).
Similarly, the introduction states [67-72] that the authors study the impact of COVID-19 distancing on the lives of medical students in terms of depression/anxiety and other behaviors. However, the analysis is merely descriptive and should be stated as such. I would not use the term “impact” as the authors are not evaluating any causal effect of social distancing policies on the lives of students. The authors provide descriptive evidence and associations.
Overall, my suggestion is to determine which are the main goals of the paper and the corresponding outcomes the authors are interested in studying and reporting and be consistent in their description. Once identified, the results section could also be re-organized with sub-sections to follow that.
- Survey
The article lacks details on the survey
- The description of the variables gathered in the survey is very long [90-100], but the survey only took 10 minutes. Why? How many questions were in the survey in total? What is the range of time?
- [86] The authors state: We used Wenjuanxing (Changsha Haoxing Information Technology Co., Ltd., China) software for the informed consent process and the online survey questionnaire.
- For a reader who is not knowledgeable of this software, more details should be provided. How does the software work? What consent does entail? Provide written consent?
- More importantly, is the survey in person? By phone? By the internet/email?
- Methods
The article also lacks important details on the methods. Overall, the authors tried to be concise, but more precision in the description of the methods is fundamental. The methods should include details on how the variable used in the analysis have been constructed.
Here some examples of things which are missing:
- How is the participant score constructed? How the cut-off has been chosen [108-109]
- More details should be provided for the HL test [115] and path analysis [16].
- Goodness-of-fit indices
In terms of knowledge, which answers have been defined as correct? The reader does not know about it without looking at Supplementary Table 1. Related to this, it is not obvious to me which answer is the correct one. The one question that I am concerned about is “which are susceptible groups for COVID-19?”. I would have answered the elderly (and in fact, 30% of respondents reported that, compared to other “wrong” answers. The answers really depend on how the question was asked, its translation in Chinese, etc. More discussion on the questions and the construction of correct answers is fundamental.
Potentially the method section could be separated in:
SAMPLE including Participants, recruitment
MAIN VARIABLES FOR ANALYSIS (or similar) including covariates and outcomes. In this section, a description of the constructed variables for the analysis should be included.
For people unfamiliar with depression and anxiety scale, I would add those in Appendix and/or briefly explain them in the methods.
- Results
Overall, the article lacks a more detailed description of the sample and summary statistics, while a lot of the discussion adds details on the data which were not described so carefully in the main text. I would consider:
- Incorporate row [122-123] in a new descriptive section
- Reduce the description in the discussion (paragraph 2-3-4) and move them in the main text. The discussion should interpret the main results and explain why certain results are found, but there should not be a detailed description of the results again. Only the main take-aways should be described.
In addition, the results section now reads as a mere description of the tables. There should be a broader description of the purpose of each piece of analysis, and how that relates to the goals of the paper. A better introduction of each table/figure should be provided relating it to the purposes as suggested above (1).
Figure 1 describes the differences between Beijing and Wuhan. I do understand what the authors are testing, that is whether the students’ knowledge and behavior differ in the epicenter vs another area. However, the hypothesis they plan to test should be explained well ahead in the methods or even in the introduction so that the reader is not surprised when one of the main results is related to this heterogenous descriptive evidence. Was the survey implemented in these two cities for a specific reason? If yes, then a better explanation should be included in the methods.
Table 3: Sex (ref= female) it means you are comparing MALE vs FEMALE if the reference is female. However, the description of the results is the opposite (women students are more likely to be depressed). Which one is correct? Maybe you should just re-label the reference group or just replace it with FEMALE. Similarly for University/site (ref= CCMU in Beijing).
- Terminology
The authors have some discussion on the use of terminology (social vs physical distancing) [58-59]. At the end of the day, distancing requires a physical separation between individuals in a social setting. To me, they are synonymous. I would suggest cutting row 58-59, which is an opinion and simply decide which term you prefer and use. As of now, you use “social distancing” in the title, distancing only in other parts of the text, etc. Pick one and be consistent.
- Discussion
I think the data presented are novel and the results very interesting. I would push this more, describing the important role that medical students play in the future health labor force. Some hints are provided in the discussion, but a broader context in the introduction and a more effective discussion would be beneficial for the article. The policy relevance of these results should be further described. The novel data on students and the results are the strengths of the paper.
Minor comments:
- …. conducted from February 4 TO 12, 2020 [23]
- Be consistent: negative thoughts/actions OR negative thoughts or actions [33-34]
- Update paragraph with the most recent figures [49-50]
- UNESCO estimates that nationwide closures have affected more than 90% of students in the world, with local closures in other countries affecting even more [5] Are these closures of school? If yes, please add that.
- Our instrument was based on social-cognitive theory and the Theory of Planned Behavior and we measured knowledge, attitude, belief and practice (KABP), enhanced with psychological assessments: add references and no capital letters
- independent variables and covariates are the same thing [114]
- Figure 1: Remove “caption” and revise the sentence “Using bar charts presents the distribution … “the subject should be Figure or bar charts present…
Author Response
Reviewer #3
- Purpose of the paper/outcomes
Both the abstract and the introduction describes the plan for the analysis, but there is inconsistency about the goals of the analysis. For example, the abstract describes (in the methods) the descriptive part, but not the assessment of the associations between covariates and depression/anxiety (even though results on those are described in results).
Response: Thank you for the critical comments. We have added the assessment of the associations into the abstract and make the context/principal findings consistent between the paper and the abstract.
Similarly, the introduction states [67-72] that the authors study the impact of COVID-19 distancing on the lives of medical students in terms of depression/anxiety and other behaviors. However, the analysis is merely descriptive and should be stated as such. I would not use the term "impact" as the authors are not evaluating any causal effect of social distancing policies on the lives of students. The authors provide descriptive evidence and associations.
Response: Thank you for the critical comments and we changed “impact” to “association”, a more appropriate description.
- Overall, my suggestion is to determine which are the main goals of the paper and the corresponding outcomes the authors are interested in studying and reporting and be consistent in their description. Once identified, the results section could also be re-organized with sub-sections to follow that.
Response: Thanks for the suggestion and helpful comments. We have added titles for sub-sections to elaborate the results, something that we agree make it more reader-friendly. We have changed the wording sequence and mention anxiety disorders ahead of depression.
4.Survey
The article lacks details on the survey
The description of the variables gathered in the survey is very long [90-100], but the survey only took 10 minutes. Why? How many questions were in the survey in total? What is the range of time?
Response: Most questions were one-choice questions, brief, and quick/easy to answer. We noted that only 300 seconds were needed if respondents reported no anxiety disorder and depression before the questionnaire was uploaded on-line. The average time for all respondents and 95%CI are 587.5 (399.0, 627.5) seconds, range (150- 2914 seconds). only 5 respondents exceeding 1000 seconds for their responses. We had mentioned that there are 84 questions in the survey.
5.[86] The authors state: We used Wenjuanxing (Changsha Haoxing Information Technology Co., Ltd., China) software for the informed consent process and the online survey questionnaire. For a reader who is not knowledgeable of this software, more details should be provided. How does the software work? What consent does entail? Provide written consent? More importantly, is the survey in person? By phone? Bythe internet/email?
Response: Thank you for asking to clarify this. The software is like SurveyMonkey®. The participants received a page of introduction of the details about the workflow of Wenjuanxing® and the study in the written consent. They choose voluntarily whether to attend. If they choose to agree to participate, they started the survey. The survey was conducted on the internet via swiping an QR code of the survey link using WeChat®, a social media App, like a combination of Facebook® and Twitter®. Both WeChat® and Wenjuanxing® are extremely popular in daily life of the students in China universities; they were used for class management and opinions are surveyed routinely. Once the participants agreed to participate in the study and swiped their QR code, the survey proceeded step by step, individually, self-administrated, and online. We have clarified this in the text.
Methods
The article also lacks important details on the methods. Overall, the authors tried to be concise, but more precision in the description of the methods is fundamental. The methods should include details on how the variable used in the analysis have been constructed.
Here some examples of things which are missing:
How is the participant score constructed? How the cut-off has been chosen [108-109]
Response: Thank you for asking for clarification. The score and cut-off were based on the provisions of the scales and we have provided the references of the scales in the methods section, lines 114-116, page 5.
7.More details should be provided for the HL test [115] and path analysis [16].
Response: Thank you for the critical comments and we strengthened the description of HL test and path analysis.
8.Goodness-of-fit indices
Response: We now specifically explain the goodness of fit indices, including the types and best values with their meanings.
9.In terms of knowledge, which answers have been defined as correct? The reader does not know about it without looking at Supplementary Table 1. Related to this, it is not obvious to me which answer is the correct one. The one question that I am concerned about is "which are susceptible groups for COVID-19?". I would have answered the elderly (and in fact, 30% of respondents reported that, compared to other "wrong" answers. The answers really depend on how the question was asked, its translation in Chinese, etc. More discussion on the questions and the construction of correct answers is fundamental.
Response: We fully agree with this. It is challenging, but since we conducted the study among the medical students and under a quite harsh social distancing environment, we think they may have much more knowledge than ordinary populations. The questions were determined according to the updated information of COVID-19 by survey time. For an emerging disease, our understanding of the COVID-19 keeps evolving and we acknowledge it in the limitation about selection of the questions and determining the correct answers.
For people unfamiliar with depression and anxiety scale, I would add those in Appendix and/or briefly explain them in the methods.
Response: Thank you; we have briefly introduced the two scales in the methods and added references to save the space.
11.Results
Overall, the article lacks a more detailed description of the sample and summary statistics, while a lot of the discussion adds details on the data which were not described so carefully in the main text. I would consider:
Incorporate row [122-123] in a new descriptive section
Response: Thank you and we incorporated these rows in a new section.
- Reduce the description in the discussion (paragraph 2-3-4) and move them in the main text. The discussion should interpret the main results and explain why certain results are found, but there should not be a detailed description of the results again. Only the main take-aways should be described.
Response: Thank you for the comments and helpful suggestion. We condensed the results and reduced repeating the numbers. However, since we are reporting the findings in a special population, we want to highlight key information for convenience of comparisons.
In addition, the results section now reads as a mere description of the tables. There should be a broader description of the purpose of each piece of analysis, and how that relates to the goals of the paper. A better introduction of each table/figure should be provided relating it to the purposes as suggested above (1).
Response: Thank you for these helpful comments; we have added a title for each result section to make the results better organized.
Figure 1 describes the differences between Beijing and Wuhan. I do understand what the authors are testing, that is whether the students' knowledge and behavior differ in the epicenter vs another area. However, the hypothesis they plan to test should be explained well ahead in the methods or even in the introduction so that the reader is not surprised when one of the main results is related to this heterogenous descriptive evidence. Was the survey implemented in these two cities for a specific reason? If yes, then a better explanation should be included in the methods.
Response: Thank you. We have strengthened the hypothesis in the introduction by adding a new last sentence of this section and emphasized it in the methods. The survey was implemented in Beijing and Wuhan to compare the differences between the capital (mild epidemic) and the pandemic epicenter.
15.Table 3: Sex (ref= female) it means you are comparing MALE vs FEMALE if the reference is female. However, the description of the results is the opposite (women students are more likely to be depressed). Which one is correct? Maybe you should just re-label the reference group or just replace it with FEMALE. Similarly for University/site (ref= CCMU in Beijing).
Response: Thank you for carefully reading and we forgot to update the information after switching the referent population and have corrected this now.
Terminology
The authors have some discussion on the use of terminology (social vs physical distancing) [58-59]. At the end of the day, distancing requires a physical separation between individuals in a social setting. To me, they are synonymous. I would suggest cutting row 58-59, which is an opinion and simply decide which term you prefer and use. As of now, you use "social distancing" in the title, distancing only in other parts of the text, etc. Pick one and be consistent.
Response: Thank you and we revised the sentences to use the word “social distancing” consistently.
Discussion
I think the data presented are novel and the results very interesting. I would push this more, describing the important role that play in the future health labor force. Some hints are provided in the discussion, but a broader context in the introduction and a more effective discussion would be beneficial for the article. The policy relevance of these results should be further described. The novel data on students and the results are the strengths of the paper.
Response: Thank you for the comments and we strengthened the significance of this part both in the sections of introduction and discussion “In the early response in epicenter in China, as well other countries, the sudden outbreak overwhelmed health professional’s preparedness in terms of personal protective equipment (PPE) shortage and psychological readiness”, “The findings underscored our study’s significance that the health reservoir suffered from the psychological stress and needed attention.”
- Minor comments:
- .... conducted from February 4 TO 12, 2020 [23]
Response: Thank you for the reminder and we have revised the sentence.
- Be consistent: negative thoughts/actions OR negative thoughts or actions [33-34]
Response: Thank you for the reminder; we have checked the full text to be consistent as “negative thoughts or actions”.
- Update paragraph with the most recent figures [49-50]
Response: We have updated with the most recent figures and have added a reference.
- UNESCO estimates that nationwide closures have affected more than 90% of students in the world, with local closures in other countries affecting even more [5] Are these closures of school? If yes, please add that.
Response: Thank you for the helpful comments and we apologize for some ambiguities in the expression of this sentence and we now reworded it as “UNESCO estimates that nationwide closures in more than 160 countries have affected over 87% of students in the world, with wider closures affecting even more”.
- Our instrument was based on social-cognitive theory and the Theory of Planned Behavior and we measured knowledge, attitude, belief and practice (KABP), enhanced with psychological assessments: add references and no capital letters
Response: Thank you; we added references.
- independent variables and covariates are the same thing [114]
Response: Thank you for the comments. We specify them differently because that in regression analyses, independent variables (i.e., the regressors) are sometimes called covariates. Used in this context, covariates are of primary interest. In most other circumstances, however, covariates are of no primary interest compared with the independent variables. We want to highlight some variables as main observation and others are less interested. [ref : ENCYCLOPEDIA, Edited by: Neil J. Salkind, Published: 2010 DOI: https://dx.doi.org/10.4135/9781412961288.n85]
- Figure 1: Remove "caption" and revise the sentence "Using bar charts presents the distribution ... "the subject should be Figure or bar charts present...
Response: We revised the caption accordingly.
Round 2
Reviewer 3 Report
Thank you for addressing all comments.